🔓 | **Open Peer Review** | Bacteriology | Methods and Protocols

# New PALM-compatible integration vectors for use in the Gram-positive model bacterium *Bacillus subtilis*

Ipek Altinoglu,[1] Rut Carballido-Lopez[1]

**ABSTRACT**  Improvements in super-resolution and single-molecule techniques, along with the development of new fluorescent proteins and labeling methods, have allowed super-resolution imaging of bacterial cells. Cloning vectors are important tools for engineering fluorescent fusions and perform efficient labeling. Here, we report the construction of four photoactivated localization microscopy (PALM)-compatible integration plasmids for the Gram-positive model organism *Bacillus subtilis*. These plasmids carry genes encoding either the photoswitchable green fluorescent protein dronPA or the photoactivatable red fluorescent protein PAmCherry1, codon-optimized or not for expression in *B. subtilis*. For fast and flexible cloning, multiple cloning sites were added at both the C-terminal and the N-terminal ends of the fluorescent protein genes. The plasmids replicate in *Escherichia coli* and allow integration at the ectopic *amyE* or *thrC* loci of *B. subtilis* via double homologous recombination, for stable chromosomal insertions of single copy number *dronPA* and *PAmCherry1* fusions, respectively. Two-color imaging is accessible with the simultaneous use of both vectors. Insertion of the LacI repressor gene under control of a constitutive promoter in each plasmid yielded four derivative vectors that, combined with an array of *lacO* operator sites, allow fluorescent repressor-operator system localization studies. We demonstrated the effective photoactivation of the LacI-dronPA and LacI-PAmCherry1 fusions, and used them to report with nanoscale precision bacteriophage SPP1 DNA within infected *B. subtilis* cells, both live and fixed, as proof of concept. Our integration vectors provide a convenient and versatile workflow for qualitative and quantitative, single- and dual-color PALM studies in *B. subtilis*.

**IMPORTANCE**  Super-resolution microscopy techniques allow localization of proteins and cellular components in prokaryotic and eukaryotic cells with unprecedented spatial resolution. Plasmids remain a powerful approach to clone fluorescent protein fusions in bacterial cells. In the current work, we expanded the toolbox of vectors available to engineer the Gram-positive model organism *Bacillus subtilis* for PALM studies. Four integrative vectors in total, two carrying the gene encoding the photoswitchable green fluorescent protein dronPA and two carrying the gene encoding the photoactivatable red fluorescent protein PAmCherry1, were constructed and tested by generating translational fusions to the LacI repressor. The LacI fluorescent fusions successfully reported the subcellular localization of viral DNA in infected *B. subtilis* cells, either live or upon fixation, by PALM. Our dronPA and PAmCherry1 integration vectors expand the genetic toolbox for single-molecule localization microscopy studies in *B. subtilis*.

**KEYWORDS**  *Bacillus subtilis*, genetic tools, PALM, integration vector, dronPA, PAmCherry, LacI/*lacO* FROS, bacteriophage SPP1

Over the last two decades, fluorescence light microscopy has become a key tool for understanding the cellular and subcellular organization of bacterial cells (1, 2).

**Peer Reviewer** Ana Oliveira Paiva, CNRS-I2BC, Gif-sur-Yvette, France

Address correspondence to Rut Carballido-Lopez, rut.carballido-lopez@inrae.fr.

The authors declare no conflict of interest.

See the funding table on p. 16.

Despite the low-cost and rapid solutions obtained with conventional light microscopes, their resolution remains limited by the diffraction of light (Abbe's diffraction limit), typically to ~200–300 nm laterally and to ~500–700 nm axially in biological specimens (3). Super-resolution (SR) microscopy techniques allow researchers to overcome this limitation (4–8) and have become available for prokaryotic cell biology studies over the last decade. They enable the resolution of subcellular structures with nanoscopic precision and provide a better understanding of the relative positions of labeled proteins, DNA, RNA, as well as their interactions and dynamic localizations. SR techniques can be classified in two groups depending on their working principle: deterministic or stochastic. The primary distinction between these two groups lies in the label-ing method. Deterministic SR techniques, such as structured illumination microscopy (SIM) and stimulated emission depletion microscopy, use conventional fluorophores [fluorescent proteins (FPs) or chemical dyes] that exhibit a non-linear response to the excitation used in order to enhance resolution (5, 6). In contrast, stochastic SR techniques require specific fluorophores that, when activated with the appropriate laser, allow for the stochastic switching on and off of single-molecule fluorescence signals over time, enabling the so-called single-molecule localization microscopy (SMLM). Techniques based on SMLM include stochastic optical reconstruction microscopy, point accumu-lation in nanoscale topography, and photoactivated localization microscopy (PALM) (4, 7, 8). In PALM, specific fluorophores are used to activate and image single mole-cules, allowing quantitative analysis and fine-scale localizations with lateral resolution down to ~10 to 30 nm (9). Three types of FPs are commonly used for PALM imag-ing: (i) photoactivatable (PA), which emit light upon activation with ultraviolet (UV) light [e.g., PA-GFP and PAmCherry1 (10, 11)]; (ii) photoconvertible, which irreversibly change their emission spectrum (generally form green to red) upon activation with UV light [e.g., mEos3.2 and mMaple3 (12, 13)], or (iii) photoswitchable, which repeatedly switch between non-fluorescent ("dark") and fluorescent states upon UV activation, until photobleaching [e.g., dronPA (14)]. Using the appropriate fluorophores and assessing the expression levels, the functionality and the stability of FP fusions is crucial in PALM imaging. The FP tag or overexpression of the FP fusion can affect the function and/or the organization of the protein of interest. Oligomerization of FPs expressed at high concentration can also lead to artifactual localization patterns and clusters, which prompted for the development of monomeric FPs (15). Furthermore, export of bacte-rial envelope proteins in an unfolded conformation via the general bacterial secretion system, prompted for the use of FPs with enhanced folding and chromophore formation (maturing) kinetics and well as increased stability, such as superfolding derivatives of GFP (16), for periplasmic and extracellular protein localization studies (17). Finally, codon optimization of the gene sequences encoding the FPs can enhance their expression and brightness in heterologous bacterial systems.

*Bacillus subtilis* is a rod-shaped, aerobic soil bacterium (18). To date, it remains the best-characterized Gram-positive bacterium for several reasons. These include its high genetic tractability, ease of cultivation and manipulation in laboratory conditions, ability to survive in extreme conditions by cellular differentiation into highly resistant spores, and efficiency in producing enzymes with industrial applications (19). For decades, *B. subtilis* has served as model organism for cell biology and subcellular organization studies in bacteria. It has played a crucial role in advancing our understanding of fundamental cellular processes such as chromosome segregation and DNA replication, cell cycle regulation, cell morphogenesis, biofilm formation, sporulation, viral infection, and antibiotic resistance, among others (20–22). The ease of genetic manipulation of *B. subtilis* results from the combination of its natural competence for genetic transforma-tion and its highly efficient homologous recombination. These features allow integrative vectors instead of replicative vectors to be commonly used to introduce synthetic DNA, which offers several advantages, notably, stable maintenance (even in the absence of selective pressure) when inserted in the chromosome via double homologous recom-bination and single copy number of the integrated genes, which can be important

**TABLE 1** Bacterial strains and phage used in this study

| Strain name | Strain type | Description | Reference |
|---|---|---|---|
| Bacterial strains | | | |
| DH5α | *E.coli* | Competent cells | Lab stock |
| ECRCL51 | *E.coli* | DH5α/pDG364 | Lab stock |
| ECRCL67 | *E.coli* | DH5α/pDG1664 | Lab stock |
| ECRCL0226 | *E.coli* | DH5α/pIA019 | Eurofins |
| ECRCL0227 | *E.coli* | DH5α/pIA020 | Eurofins |
| ECRCL0287 | *E.coli* | DH5α/pIA021 | This study |
| ECRCL0289 | *E.coli* | DH5α/pIA023 | This study |
| ECRCL0290 | *E.coli* | DH5α/pIA025 | This study |
| ECRCL0292 | *E.coli* | DH5α/pIA027 | This study |
| ECRCL0293 | *E.coli* | DH5α/pIA028 | This study |
| ECRCL0295 | *E.coli* | DH5α/pIA030 | This study |
| ECRCL0297 | *E.coli* | DH5α/pIA025 | This study |
| ECRCL0299 | *E.coli* | DH5α/pIA034 | This study |
| ECRCL0300 | *E.coli* | DH5α/pIA033 | This study |
| ECRCL0301 | *E.coli* | DH5α/pIA035 | This study |
| ECRCL0302 | *E.coli* | DH5α/pIA036 | This study |
| YB886 | *B. subtilis* | *amyE trpC2 metB5 xin-1 attSPβ* | (31) |
| GSY10004 | *B. subtilis* | YB886, *thrC::(P$_{pen}$-lacIΔ11-mcherry mls erm2)* | (32) |
| RCL0937 | *B. subtilis* | YB886, *amyE::(P$_{pen}$-lacIΔ11-dronPA mls cm)* | This study |
| RCL0938 | *B. subtilis* | YB886, *thrC::(P$_{pen}$-lacIΔ11-PAmCherry1 mls erm2)* | This study |
| RCL0939 | *B. subtilis* | YB886, *thrC::(P$_{pen}$-lacIΔ11-bsPAmCherry1 mls erm2)* | This study |
| RCL0941 | *B. subtilis* | YB886, *amyE::(P$_{pen}$-lacIΔ11-bsdronPA mls cm)* | This study |
| Phage | | | |
| *lacO64* | SPP1 | *SPP1delX110lacO64* | (30) |

in terms of regulation (23–26). Vectors (here restricted to plasmid vectors) have been therefore extensively modified to meet custom needs and applications. Four non-essential integration loci, *amyE*, *thrC*, *sacA*, and *lacA,* are traditionally used for ectopic insertion of genetic constructions in the *B. subtilis* chromosome.

Here, we developed and tested four integrative vectors: two for insertion into the *thrC* locus, enabling the expression of fusions to the photoactivatable monomeric red FP PAmCherry1, encoded either by the original gene sequence or by a codon-optimized version for expression in *B. subtilis* (*bsPAmCherry1*), and two for insertion into the *amyE* locus, to express fusions to the photoswitchable green FP dronPA, encoded by either the original *dronPA* gene sequence or a codon-optimized variant for expression in *B. subtilis* (*bsdronPA*). We confirmed the integration of the vectors and the functionality of the fluorophores in both live and fixed cells using the commonly used fluorescent repressor operator system (FROS) LacI-*lacO* (27, 28). Bacteriophage SPP1 DNA, engineered to carry an array of ~64 *lacO* operator sites inserted in its genome, was previously visualized during infection of *B. subtilis* cells producing a chromosomally encoded LacI-mCherry fusion (29, 30). We constructed four derivative vectors enabling the integration of *lacI-PAmCherry1* or *lacI-dronPA*, or their codon-optimized variants, under control of a constitutive promoter in the *B. subtilis* chromosome. Finally, we demonstrated the versatility of these vectors by using them to visualize the nanoscale localization of bacteriophage SPP1 DNA in infected cells using single-color PALM.

## MATERIALS AND METHODS

### Bacterial strains and growth conditions

The bacterial strains and the phage used in this study are listed in Table 1. *B. subtilis* YB886 (31) and *E. coli* DH5α were used as wild-type strains. Unless specified otherwise,

bacterial strains were grown in lysogeny broth (LB) or LB agar (1.5%). Moreover, when appropriate, antibiotics were used at the following concentrations: ampicillin 100 µg/mL, chloramphenicol 25 µg/mL, and erythromycin 30 µg/mL for *E. coli*; and chloramphenicol 5 µg/mL, erythromycin 0.5 µg/mL, lincomycin 12.5 µg/mL, and streptomycin 100 µg/mL for *B. subtilis*.

## DNA manipulation and plasmids construction

### General cloning procedures

Polymerase chain reaction (PCR) amplifications were carried out using the Phusion polymerase. Restriction enzyme digestions were performed using New England Biolabs (NEB) high-fidelity enzymes and their corresponding protocols, as suggested by the supplier. All plasmids were constructed by isothermal assembly (33) using a minimum of 20-bp overlapping regions between DNA fragments and a custom-made kit. Plasmid selection and verification after transformation were performed using the OneTaq 2× Master Mix with Standard Buffer (NEB). Plasmids were purified using the Qiagen plasmid purification kit according to the manufacturer's protocol.

The plasmids and oligonucleotides used in this study are listed in Tables 2 and 3, respectively.

### Construction of multiple cloning sites

The overlapping oligonucleotides ITAup (45 bp) and ITAdown (45 bp) were annealed by DNA polymerase reaction. The resulting double-stranded DNA is called multiple cloning site (MCS) (Fig. 1B).

### Construction of empty integration vectors pIA021 and pIA023

The MCS was amplified using primers oIA001 and oIA002 and cloned by Gibson Assembly into pDG364 (25) (*amyE::cm*) digested with *EcoRI* and *BamHI*, resulting in plasmid pIA021. MCS amplified with primers oIA003 and oIA004 was cloned by Gibson Assembly into pDG1664 (23) (*thrC::erm2*) digested by *EcoRI* and *BamHI*, resulting in primer pIA023.

### Construction of plasmids pIA027, pIA034, pIA028, and pIA033 containing PALM-compatible fluorescent proteins

The PALM-compatible FPs dronPA (in plasmid pEYY99) and PAmCherry1 (in plasmid pEYY133) were kindly provided by Dr. Yoshiharu Yamaichi (34). The *dronPA* and *PAmCherry1* genes were sequenced for confirmation. The *dronPA* and *PAmCherry1* gene sequences codon-optimized for *B. subtilis* (*bsdronPA* and *bsPAmCherry1,* respectively) were synthesized and cloned into a pEX backbone plasmid by Eurofins Genomics, giving pIA019 (pEX-A128-bsdronPA) and pIA020 (pEX-A128-bsPAmCherry1). The *dronPA* sequence was amplified from pEYY99 using primers oIA010 and oIA047, then cloned into pIA021 digested with *BamHI* and *SbfI*, resulting in plasmid pIA027, *amyE::(dronPA cm)*. The *PAmCherry1* sequence was amplified from pEYY133 using primers oIA008 and oIA048, then cloned into pIA023 digested with *BamHI* and *SphI*, resulting in plasmid pIA028, *thrC::(PAmCherry1 erm2)*. *bsdronPA* was amplified from pIA019 using primers oIA054 and oIA055, then cloned into pIA021 digested with *BamHI* and *HindIII*, giving plasmid pIA034, *amyE::(bsdronPA cm). bsPAmCherry1* was amplified from pIA020 using primers oIA056 and oIA057, then cloned into pIA023 digested with *BamHI* and *HindIII*, giving plasmid pIA033, *thrC::(bsPamCherry1 erm2)*. The newly constructed plasmids (pIA027, pIA028, pIA034, and pIA033) re-created two additional restriction sites, *EagI* and *NotI*, next to the MCS via primer encoding. All plasmids were verified by Sanger DNA sequencing (GATC Biotech/Eurofins Genomics, Ebersberg, Germany). Plasmids *pIA027*, *pIA034*, *pIA028*, and *pIA033* were fully sequenced and deposited at Addgene with catalog ID nos. 200418, 200421, 200419, and 200420, respectively.

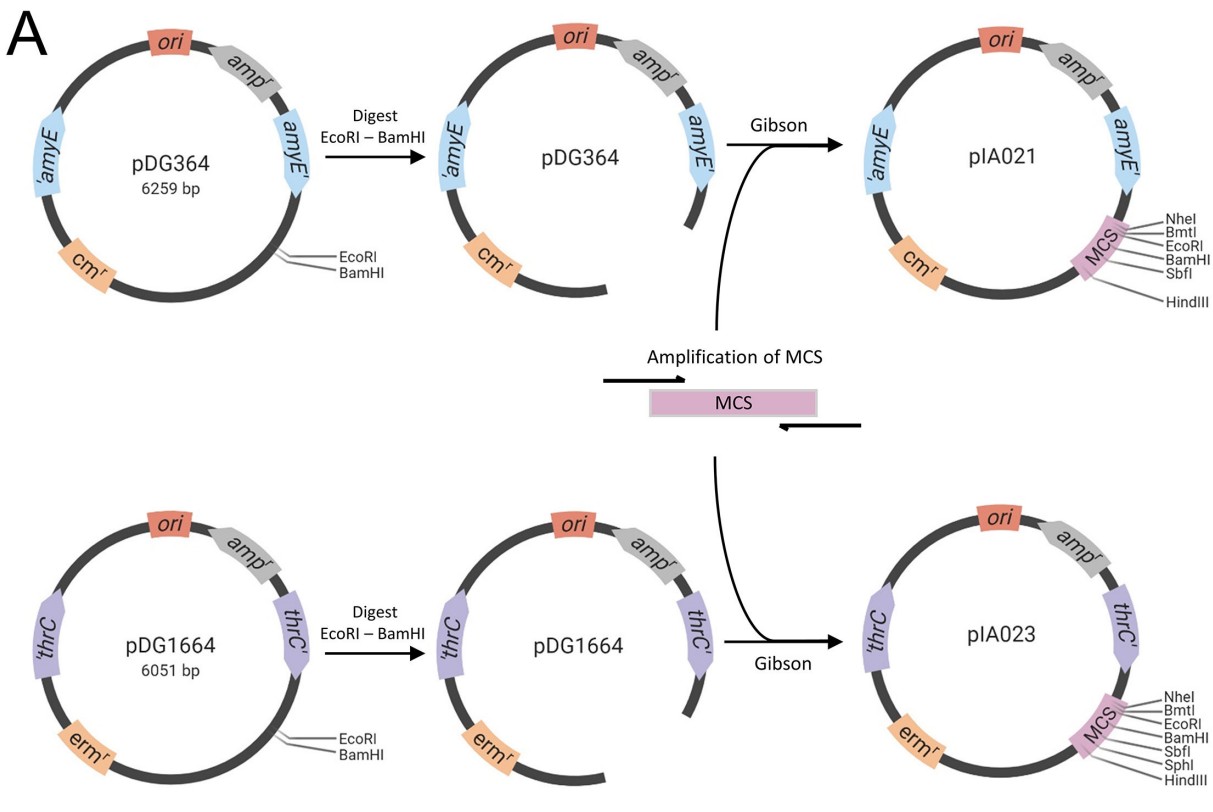

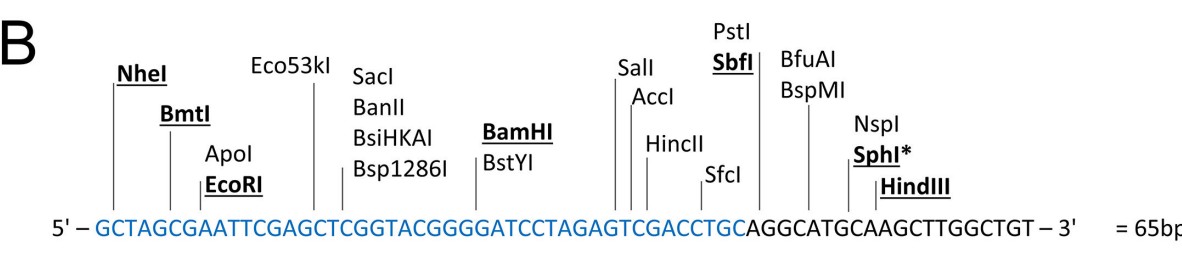

**FIG 1** Construction of the empty integrative plasmids pIA021 and pIA023 for efficient cloning. (A) Plasmids pDG364 and pDG1664 were used as backbones. The replication origin (*ori*) and ampicillin resistance cassette (amp^r) for vector verification and propagation in *E. coli* are indicated in light red and gray, respectively. The *B. subtilis* genomic integration sites are shown in blue for *amyE* and in purple for *thrC*. The chloramphenicol and erythromycin resistance cassettes (cm^r and erm^r) for selection in *B. subtilis* are indicated in orange. The multiple cloning site (MCS) is shown in pink. Arrows indicate the plasmid construction process (see the Materials and Methods section for details). The MCS was inserted into the pDG364 and pDG1664 empty plasmids by Gibson Assembly. (B) The MCS was engineered using two overlapping oligonucleotides of 45 bp each (in blue) that were annealed to generate a double-stranded DNA fragment of 65 bp containing 22 restriction sites. Underlined in bold are the restriction sites unique in the new plasmids pIA021 and pIA023. The star indicates the *SphI* site, which is only unique in the MCS for plasmid pIA023.

**TABLE 2** List of plasmids

| Plasmid number | Description | Reference |
|---|---|---|
| pDG364 | *amyE::cm* | (23) |
| pDG1664 | *thrC::erm2 (thrC Amp$^r$ Erm$^r$ Spec$^r$)* | (25) |
| pPT300 | *thrC::(P$_{pen}$-lacIΔ11-mcherry mls erm)* | (32) |
| pEYY133 | Cloning vector for C-terminal *PAmCherry1* fusion | (34) |
| pEYY99 | Cloning vector for C-terminal *dronPA* fusion | (34) |
| pIA019 | *pEX-A128-bsdronPA* | This study |
| pIA020 | *pEX-A128-bsPAmCherry1* | This study |
| pIA021 | *amyE::(MCS cm)* | This study |
| pIA023 | *thrC::(MCS erm2)* | This study |
| pIA025 | *thrC::(P$_{pen}$-lacIΔ11-bsPAmCherry1 mls erm2)* | This study |
| pIA027 | *amyE::(dronPA cm)*, for *dronPA* fusions | This study |
| pIA028 | *thrC::(PAmCherry1 erm2)*, for *PAmCherry1* fusions | This study |
| pIA030 | *thrC::(P$_{pen}$-lacIΔ11-PAmCherry1 erm2)* | This study |
| pIA033 | *thrC::(bsPamCherry1 erm2)*, for *bsPAmCherry1* fusions | This study |
| pIA034 | *amyE::(bsdronPA cm)*, for *bsdronPA* fusions | This study |
| pIA035 | *amyE::(P$_{pen}$-lacIΔ11-dronPA cm)* | This study |
| pIA036 | *amyE::(P$_{pen}$-lacIΔ11-bsdronPA cm)* | This study |

## Construction of plasmids pIA035, pIA036, pIA030, and pIA025 containing PALM-compatible fluorescent fusions to LacI

Plasmid pPT300, *thrC::(Ppen-lacIΔ11-mcherry mls erm)* was kindly provided by Dr. Paulo Tavares (32). The *lacI* ORF together with the ribosomal binding site (RBS) and promoter region sequences were amplified and inserted in plasmids pIA027, pIA034, pIA028, and pIA033 as follows.

pPT300 was amplified using primers oIA027 and oIA058. The resulting fragment was cloned into pIA027 digested by *BamHI* and *NotI*, giving plasmid pIA035, *amyE::(Ppen-lacIΔ11-dronPA cm)*.

pPT300 was amplified using primers oIA027 and oIA069. The resulting fragment was cloned into pIA034 digested with *BamHI* and *NotI*, giving pIA036, *amyE::(Ppen-lacIΔ11-bsdronPA cm)*.

pPT300 was amplified using primers oIA027 and oIA028. The resulting fragment was cloned into pIA028 digested with *BamHI* and *NotI*, giving pIA030, *thrC::(Ppen-lacIΔ11-PAmCherry1 erm2)*.

pPT300 was amplified using primers oIA027 and oIA036. The resulting fragment was cloned into pIA033 digested with *BamHI* and *NotI*, giving pIA025, *thrC::(Ppen-lacIΔ11-bsPAmCherry1 mls erm2)*.

All constructs were verified by Sanger DNA sequencing (GATC Biotech/Eurofins Genomics). Plasmids *pIA035*, *pIA036*, *pIA030,* and *pIA025* were then fully sequenced and deposited at Addgene with catalog ID nos. 200426, 200425, 200424, and 200423, respectively.

### Bacterial transformation and selection

*E. coli* DH5α competent cells were used for routine cloning applications and for transformation of all plasmids at minimum 50 ng/µl. Cells harboring the plasmids were selected by plating transformants on LB agar plates containing the appropriate antibiotic and incubating them overnight at 37°C. Single colonies grown overnight on the selective medium were screened by colony PCR to determine the presence or absence of the fragment inserted in the plasmid using check primers between Check01 and Check08 in Table 3. Positive colonies were grown overnight in LB medium at 37°C with agitation (180 rpm), and plasmids were purified.

**TABLE 3** List of oligonucleotides

| Name | Sequence (5′–3′) |
| --- | --- |
| ITA_up | GCTAGCGAATTCGAGCTCGGTACGGGGATCCTAGAGTCGACCTGC |
| ITA_down | ACAGCCAAGCTTGCATGCCTGCAGGTCGACTCTAGGATCCCCGTA |
| Check01_PAmCherry_C | CCTCGTAGGGGCGGCCCTCGC |
| Check02_PAmCherry_N | GCGCCTACAACGTCAACCGC |
| Check03_dronPA_C | CCATACTCTGTTTTCCCTCG |
| Check04_dronPA_N | GCCAGACTATCACTTTGTGG |
| Check05_AmyE up | TTGCAAAACGATTCAAAACCTCTT |
| Check06_AmyE down | GCTGATTCTGACCGGGCACTTGG |
| Check07_ThrC up | ACCGGCGCTAACTTTACATGAAGG |
| Check08_ThrC down | TCATCAGTCGGCAATGTGACAGG |
| IA001 | GCGACCGGCGCTCAGGATCGCTAGCGAATTCGAGCTCGG |
| IA002 | GATAAGCTGTCAAACATGAGAATTACAGCCAAGCTTGCATGCC |
| IA003 | CCAAAAAACTGCTGCCTTCGGATCGCTAGCGAATTCGAGCTCGG |
| IA004 | CCATAACTTTAGGGTTATCGAATTACAGCCAAGCTTGCATGCC |
| IA008 | TTACAGCCAAGCTTGCATGCCTTTACTTGTACAGCTCGTCCATGCCGC |
| IA010 | GCCAAGCTTGCATGCCTGCAGGTTTACTTGGCCTGCCTCGGCAGCTC |
| IA027 | CGAATTCGAGCTCGGTACGGGGATCGGGTTATCGAATTCCCGGTGG |
| IA028 | CCTCCTCGCCCTTGCTCACGGCGGCCGCCAGCTGCATTAATGAATCGGCC |
| IA036 | CCTCCTCGCCCTTTGGACATGGCGGCCGCCAGCTGCATTAATGAATCG |
| IA047 | CGAGCTCGGTACGGGGATCCTGCGGCCGCCATGAGTGTGATTAAACCAGAC |
| IA048 | CGAGCTCGGTACGGGGATCCTGCGGCCGCCGTGAGCAAGGGCGAGGAGG |
| Ia054 | CGAGCTCGGTACGGGGATCCTGCGGCCGCCATGTCAGTCA |
| IA055 | CAAACATGAGAATTACAGCCAAGCTTGCATGCCTGCAGGTTTATTTC |
| IA056 | CGAGCTCGGTACGGGGATCCTGCGGCCGCCATGTCCAAAGGCG |
| IA057 | AGGGTTATCGAATTACAGCCAAGCTTGCATGCCTTTACTTATAC |
| IA058 | GGTTTAATCACACTCATGGCGGCCGCCAGCTGCATTAATGAATCGG |

*B. subtilis* YB886 competent cells were used for integration of the FROS vectors pIA035, pIA036, pIA030, and pIA025. Competent cells were prepared by a two-step starvation protocol as previously described (35). At least 1 µg of circular plasmid was added to 500 µL of competent cells. Cells were plated onto appropriate selective media and incubated overnight at 37°C. Single colonies grown overnight were patched onto replica plates for replication and isolation prior to use in colony PCR to amplify the *amyE* or *thrC* regions.

## Phage infection and sample preparation for fluorescence microscopy

*B. subtilis* strains were plated with the appropriate antibiotic selection and incubated overnight at 37°C to isolate single colonies. A single colony was grown overnight in LB medium at 30°C with agitation (180 rpm). Overnight cultures were diluted 1:100 in fresh LB medium and grown at 37°C with agitation (180 rpm). When cultures reached an $OD_{600\ nm}$ of 0.8, they were supplemented with 10 mM $CaCl_2$ and infected with phage SPP1 (strain SPP1*delX110lacO$_{64}$*) at an input multiplicity of 1. Infected cells were further incubated at 37°C with orbital agitation (180 rpm) for 50 min and spotted onto an agarose pad (1% agarose with $H_2O$) mounted on a glass slide using a Gene Frame (Thermo Fisher). Cover slips (thickness no. 1.5H) and/or slides were pre-cleaned with acetone and then plasma cleaned with Harrick plasma (PDC-002-CE 230V) for 10 min at high level.

Images of live cells were acquired at 50 min post-infection (p.i.). For image acquisition of fixed cells, infected cells were fixed 50 min p.i. with fresh fixing solution (1 M $KPO_4$ pH7, 2% PFA, and 0.02% glutaraldehyde) in a 2:1 ratio (2-vol cells:1-vol fixing solution). Fixed cells were incubated for 15 min at room temperature, then 30 min in ice, washed with 100 µL of PBS and kept overnight at 4°C.

All images were acquired using a Zeiss Elyra PS1 microscope, at 37°C for live cells and at room temperature for fixed cells. The Zeiss Elyra PS1 was equipped with a ×63 (NA 1.40) and an Apo ×100 (NA 1.46) Apochromatic oil immersion objectives, coherent lasers emitting at 405 nm (50 mW), 488 nm (100 mW), and 561 nm (100 mW), an emCCD Andor iXon 897 camera, and a PCO edge sCMOS camera. The microscope and cameras were controlled by the Zen software version ZEN 2012 SP2.

## SIM imaging

SIM images (Fig. 3A,B) were taken in fields of 256 × 256 pixels using the sCMOS camera and the ×63 (NA 1.40) objective. One hundred-millisecond exposures were used for white field illumination and 50 ms exposures were used for the 561 nm laser at 50%. 34.0 µm G3 grid with 3 phases and five rotations was used for SIM imaging. Reconstruction of SIM images was made using the FairSIM (36) plug-in of Fiji (37) and the Zen 2012 SP2 image analysis software equipped for two-dimensional imaging.

## PALM imaging

PALM images were taken in fields of 512 × 512 pixels using the EMCCD camera, a ×100 (NA 1.46) objective, a HILO angle of 43.74°, and 25 ms exposures. The 488- or 561-nm lasers were used for the first 1,500 frames to bleach out the cell background. Continued activation with the 405 nm and the 488- or 561-nm lasers was used for molecule detection over 4,000 frames. The ThunderSTORM (38) plug-in of Fiji was used for molecule detection and quantification analysis. Output coordinate files include information for each detected molecule, such as x,y location, photon count, uncertainty, and sigma. The output files of ThunderSTORM were used in the Abbelight Neo Analysis software to identify and characterize clusters using DBSCAN cluster analysis, providing parameters such as the number of localizations per cluster, cluster area, cluster density, and radius of gyration (center of mass). Final super-resolution PALM images were reconstructed using the normalized Gaussian fitting method and a 15 nm pixel size. $Z$ projections were used as source image to compare the resolution improvement relative to PALM images.

## RESULTS

### Design of the integration vectors and insertion of multiple cloning sites

To generate standardized cloning plasmids for chromosomal expression of PALM-compatible FP fusions upon double cross-over homologous recombination in *B. subtilis*, we chose two well-established empty vectors, pDG364 and pDG1664, as plasmid backbones. These vectors integrate, respectively, at the *amyE* and *thrC* loci of *B. subtilis* (23, 25) (Fig. 1A). Both plasmids contain the *E. coli* origin of replication (*ori*), to maintain and propagate the foreign plasmid in *E. coli*, and the *bla* gene (indicated as AmpR in the plasmids maps), which encodes β-lactamase and thus allows the selection of the plasmid in *E. coli* with ampicillin. In addition, pDG364 and pDG1664 harbor *cm* and *erm* cassettes, respectively, for selection in *B. subtilis* using chrolamphenicol and erythromycin, respectively. We chose two different integration loci and selection markers to allow the final vectors to be combined in the same strain for multicolor imaging. To facilitate the cloning, we engineered an MCS by using two overlapping oligonucleotides of 45 bp each (Fig. 1B) that were annealed to generate a double-stranded DNA fragment of 65 bp, called MCS, which includes 22 unique restriction sites (Fig. 1B). Isothermal assembly (Gibson) (33) was used to integrate the MCS into pDG364 and pDG1662 linearized by *EcoRI* and *BamHI* digestion (Fig. 1A). The resulting circular vectors pIA021 and pIA023 possess six common unique restriction sites: *NheI*, *BmtI*, *EcoRI*, *BamHI*, *SbfI*, and *HindIII*, and pIA023 possesses one additional unique restriction site: *SphI*.

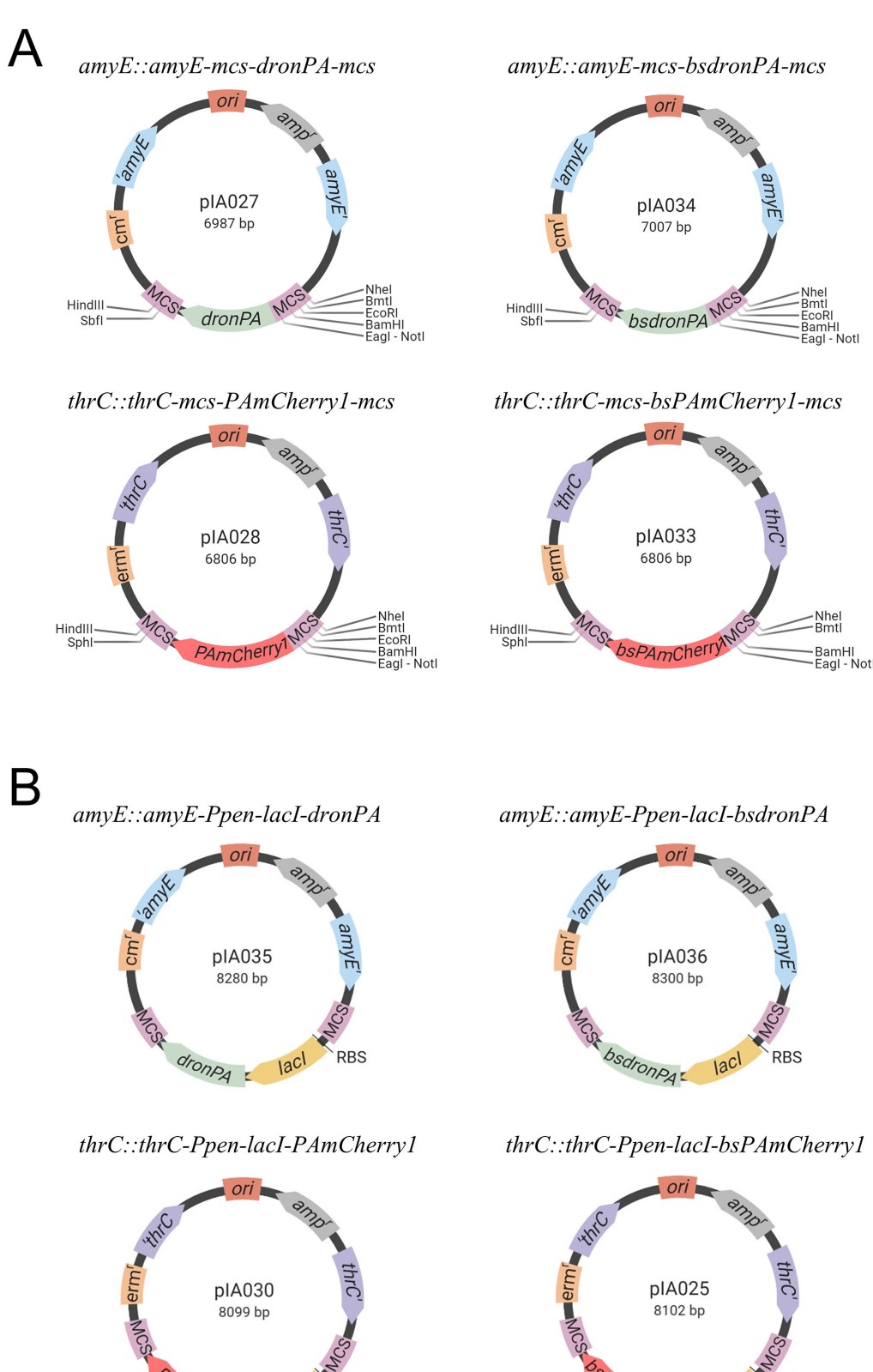

**FIG 2** PALM-compatible integration plasmids for *B. subtilis*. The replication origin (*ori*) and ampicillin resistance cassette (amp^r)
for *E. coli* are marked in light red and gray, respectively. The *B. subtilis* genomic integration sites are shown in blue for *amyE*
purple for *thrC*, and the resistance cassettes are indicated in orange. Multiple cloning sites (MCSs) are

Fig 2 (Continued)

shown in pink. The PALM-compatible fluorescent fusions are indicated in light green and magenta for *dronPA/bsdronPA* and *PAmCherry1/bsPAmCherry1*, respectively. (A) Integration plasmids carrying the genes encoding the PALM-compatible fluorescent proteins dronPA and PAmCherry1: original *dronPA* sequence, plasmid pIA027; codon-optimized sequence for *B. subtilis bsdronPA*, plasmid pIA034; original PAmCherry1 sequence, plasmid pIA028; codon-optimized sequence for *B. subtilis bsPAmCherry1*, plasmid pIA033. (B) Integration plasmids for LacI/*lacO* FROS system in *B. subtilis*. The *lacI* gene, shown in yellow, was cloned from plasmid pPT300 together with its RBS and promoter region, fused to the N-terminus of the gene encoding the PALM-compatible fluorophores; original *dronPA* sequence, plasmid pIA035; codon-optimized sequence for *B. subtilis bsdronPA*, plasmid pIA036; original PAmCherry1 sequence, plasmid pIA030; codon-optimized sequence for *B. subtilis bsPAmCherry1*, plasmid pIA025.

## New plasmids with PALM-compatible fluorescent proteins PAmCherry1 and dronPA

PALM-compatible FPs play a critical role in SMLM image acquisition and depend on factors such as their activation mechanism, photon detection, fluorophore lifetime, and oligomerization properties. In addition to considerations for single-color imaging, the selection of the right combination of FPs for multicolor PALM imaging becomes notably complex due to two main reasons: (i) the requirement for multiple laser sources (at least two, for activation and imaging) for each fluorophore, and (ii) limitations imposed by the photophysical properties of FPs, precluding photoconvertible fluorophores (e.g., mEos3.2 and mMaple3), which use three laser sources, to be combined with any other PA FPs. Therefore, significant time and effort may be required to identify the most suitable fluorophore(s) for a specific experiment. To generate plasmids compatible with two-color imaging, we first chose a suitable FP pair: the photoswitchable monomeric green fluorescent protein dronPA (14) [originally isolated from *Echinophyllia* sp. *SC22*; fluorophore excitation 503 nm, emission 518 nm (green), photoactivation with UV 405 nm] and the photoactivatable monomeric red fluorescent protein PAmCherry1 (11) [derived from mCherry from *Discosoma* sp.; fluorophore excitation 564 nm, emission 595 nm (red), photoactivation with UV 405 nm]. The dronPA and PAmCherry1 are both initially in dark state, photoactivated with 405 nm light, and spectrally distant, ensuring minimal cross-talk. Furthermore, they are monomeric and have been successfully used for dual-color membrane labeling in bacterial cells (34).

We used plasmids pIA021 and pIA023 to introduce the gene coding for the photo-switchable FP dronPA and for the photoactivatable FP PAmCherry1, respectively, into the MCS using isothermal assembly. Both the original gene sequences and their codon-optimized versions for expression in *B. subtilis* were cloned. The resulting four new plasmids are pIA027 (*amyE::amyE-mcs-dronPA-mcs*), pIA034 (*amyE::amyE-mcs-bsdronPA-mcs*), pIA028 (*thrC::thrC-mcs-PAmCherry1-mcs*), and pIA033 (*thrC::thrC-mcs-bsPAmCherry1-mcs*) (Fig. 2A; Table 2). They allow the cloning of the promoter region and the gene of interest fused to the N-terminal of the gene encoding the FP, using either isothermal assembly or classical enzyme digestion/ligation methods. Cloning of the gene of interest fused to the C-terminal of the gene encoding the FP is also possible by isothermal assembly. In this case, a two-step isothermal assembly is to be performed, which nowadays is an accessible and straightforward cloning approach. First, the stop codon of the *dronPA* or *PAmCherry1* genes must be removed, and a linker must be added between the C-terminal of FP and the gene of interest using annealing sequences. Second, the promoter of the gene of interest should be added upstream the gene encoding the FP. Pairwise combinations of these plasmids can allow simultaneous insertion at *amyE* and *thrC* of two genes of interest, fused to *dronPA* (or *bsdronPA*) and *PAmCherry1* (or *bsPAmCherry1*), respectively, for dual-color PALM imaging.

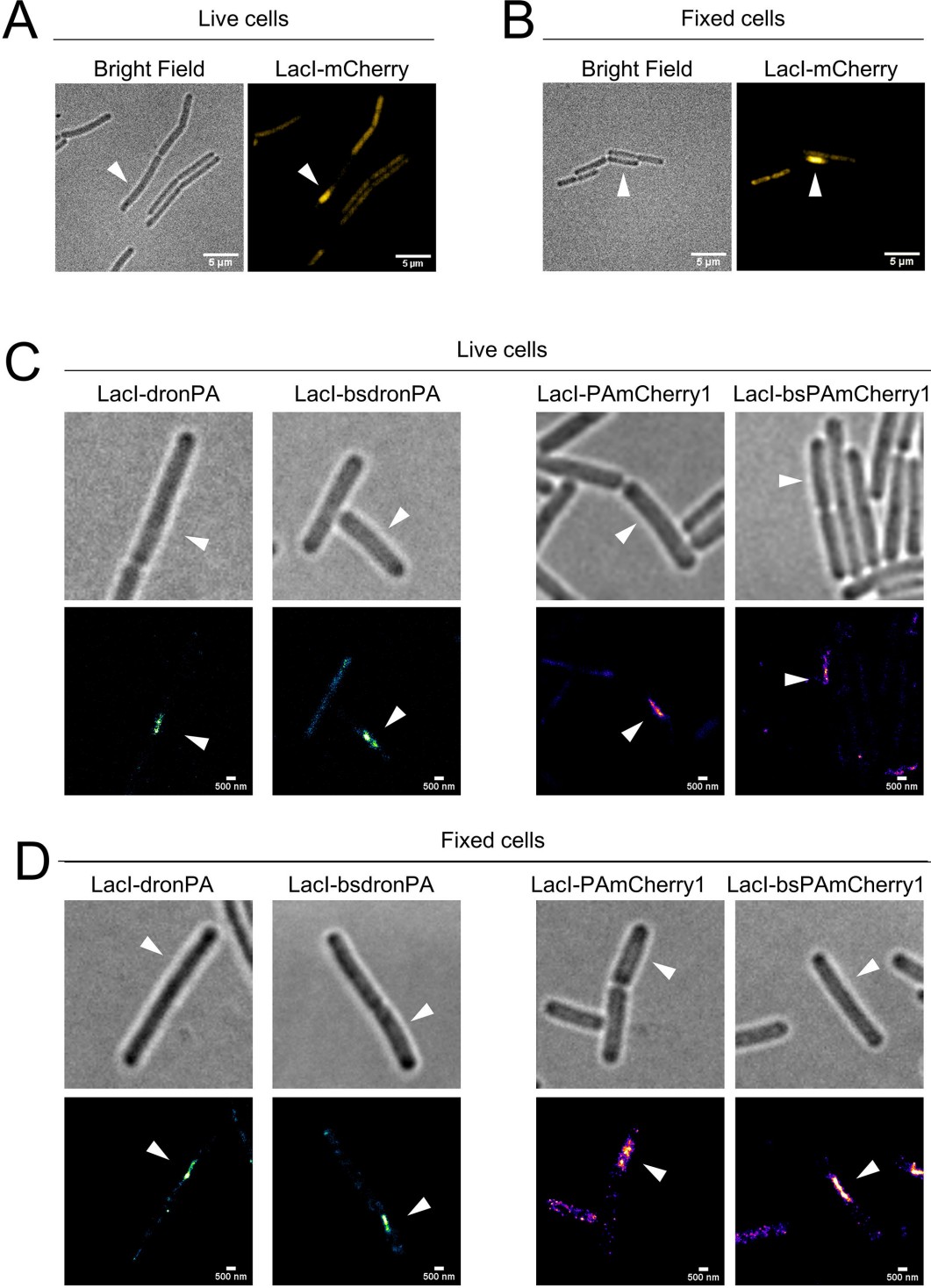

**FIG 3** Visualization of bacteriophage SPP1 DNA in infected *B. subtilis* cells. Imaging of bacteriophage SPP1 viral DNA in infected cells of *B. subtilis* cells for validation of our PALM-compatible vectors and confirmation of the methodology. *B. subtilis* cells constitutively expressing the different LacI-FP fusions from the ectopic loci were infected with phage SPP1*delX110lacO64* and imaged 50 min post-infection. In infected cells, the LacI repressor binds to the *lacO* operator repeats in the bacteriophage DNA, enabling visualization of the replicating viral DNA. White arrowheads indicate the infected cells. (A and B) LacI-mCherry localization in live (A) and fixed (B) cells of strain GSY10004, imaged by SIM. Scale bars, 5 µm. (C and D) Single-molecule localization of dronPA (strain RCL0937), bsdronPA (strain RCL0941), PAmCherry1 (strain RCL0938), and bsPAmCherry1 (strain RCL0939) fusions to LacI in live (C) and fixed (D) cells at 50 min p.i. DronPA/bsdronPA localizations are shown in green, and PAmCherry1/bsPAmCherry1 localizations are shown in magenta. Scale bars, 500 nm.

## Functionality of the plasmid-bore PALM-compatible FPs for FROS localization studies in *B. subtilis*

To confirm the functionality of our new vectors, pIA027, pIA034, pIA028, and pIA033, we used a FROS (27, 28). FROS systems are used to label specific DNA segments through binding of a fluorescently labeled repressor to repeated sequences of its cognate operator. They allow visualization of DNA loci in cells to understand genome organization and dynamics, and to determine copy numbers of genetic loci (39). Here, we chose the repressor and the operator of the lactose operon (LacI/*lacO*) to label bacteriophage SPP1 DNA in SPP1-infected *B. subtilis* cells. A similar LacI/*lacO* FROS system was previously used to investigate the subcellular localization of the viral DNA during infection using conventional epifluorescence microscopy (29, 30). Upon irreversible binding to the surface of a *B. subtilis* host cell, bacteriophage SPP1 injects its DNA into the cytoplasm of the bacterium. Inside the host cell, the viral DNA undergoes replication to produce viral proteins and assemble new virions, which are ultimately released into the medium by inducing the lysis of the host cell (32). Using a chromosomally encoded, constitutively expressed LacI-mCherry fusion and a phage engineered to carry an array of ~64 *lacO* operator sites inserted in its genome (SPP1*delX110lacO64*), the viral DNA was visualized within infected cells. This revealed that a single DNA focus that harbors viral genome replication, referred to as the viral DNA compartment, is formed and increases in size throughout the infection process (29, 30, 32). We cloned the *lacI* gene under control of the constitutive P$_{pen}$ promoter into our four PALM-compatible vectors, pIA027, pIA028, pIA033, and pIA034, fused to the N-terminus of the gene encoding the FP. To this end, the plasmids were linearized by digestion with the *BamHI* and *NotI* high-fidelity enzymes (fragment-I or "vector") and assembled with a fragment containing the P$_{pen}$ promoter, the RBS for efficient translation, and the *lacI* gene (fragment-II or "gene of interest"). The resulting four new plasmids, pIA025 (*thrC::thrC-lacI-bsPAmCherry1*), pIA030 (*thrC::thrC-lacI-PAmCherry1*), pIA035 (*amyE::amyE-lacI-dronPA*), and pIA036 (*amyE::amyE-lacI-bsdronPA*) (Fig. 2B; Table 2), were transformed into *B. subtilis*, generating strains RCL0937, RCL0938, RCL0939, and RCL0941, respectively (Table 1). These strains were then infected with phage SPP1*delX110lacO64*. The previously reported strain GSY10004 expressing LacI-mCherry was used as control of SPP1 DNA localization. This strain was imaged using SIM, which enhances lateral resolution approximately two-fold compared to conventional epifluorescence microscopy (~100 to 150 nm and ~200 to 300 nm, respectively). Mono infected cells displayed a single-phage DNA focus (Fig. 3A, arrowhead), as previously reported (29, 30). Mild fixation of cells did not affect the localization pattern of the viral DNA compartment (Fig. 3B). Infected cells expressing the LacI-PAmCherry1, LacI-bsPAmCherry1, LacI-dronPA, or LacI-bsdronPA fusions were then imaged by PALM, using HILO illumination to enhance the signal-to-noise ratio of the thin bacterial cell samples. PALM images were acquired from live cells (Fig. 3C) and after fixation (Fig. 3D). In all cases, successful localization of the bacteriophage DNA was observed. We achieved lateral localization precision (uncertainty from coordinate files) from detected molecules of ~21.8 nm for dronPA, ~24.4 nm for bsdronPA, ~23.8 nm for PAmCherry1, and ~24.2 nm for bsPAmCherry1 (Fig. 4A), indicating a remarkably similar precision between photoswitching and photoactivated fluorophores.

The final reconstructed PALM images were compared with source images for each fusion (Fig. 4). Average intensity line plots across the viral DNA compartments showed a significant reduction of their width in PALM images, confirming the enhancement of resolution (Fig. 4B). In addition to the gain in resolution, photoactivation allowed us to quantify the number of localizations. We identified single viral DNA clusters in fields containing infected and non-infected cells (Fig. 5A) and determined their centroid. We next quantified cluster area, density, and radius of gyration, as well as the number of molecules for the photoactivatable FPs and the number of localizations for the photoswitchable FPs (Fig. 5B). For photoactivatable FPs (PAmCherry1/bsPAmCherry1), each localization corresponds to one molecule, whereas for photoswitchable FPs (dronPA/bsdronPA), molecules blink, resulting in multiple localizations from a single

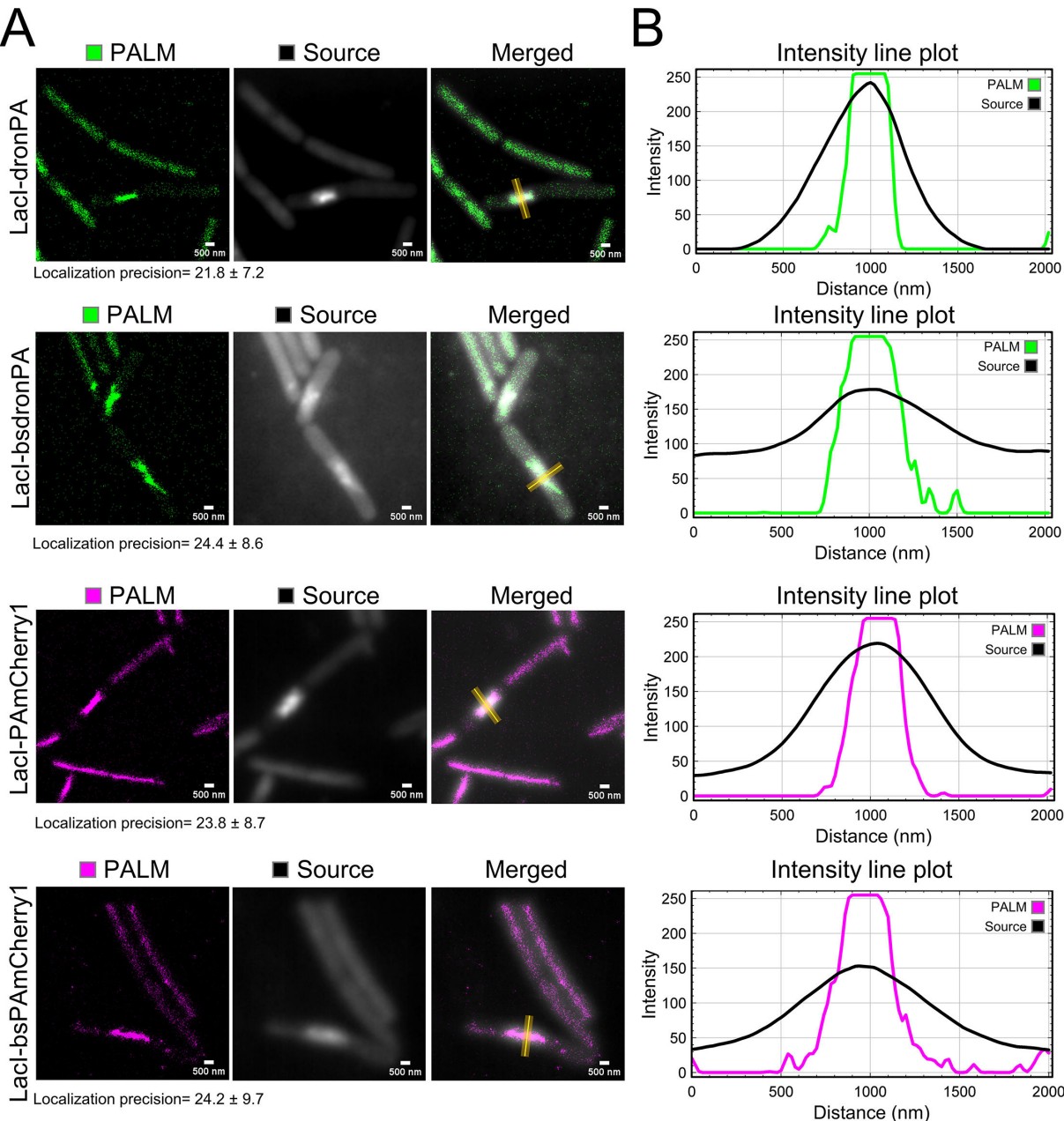

**FIG 4** Resolution improvement with PALM. (A) Super-resolution PALM and corresponding source images of the viral DNA compartment in infected *B. subtilis* cells expressing the LacI-dronPA (green) and LacI-PAmCherry1 (magenta) fusions. Cells of strains RCL0937 (LacI-dronPA), RCL0941 (LacI-bsdronPA), RCL0938 (LacI-PAmCherry1), and RCL0939 (LacI-bsPAmCherry1) were monoinfected with phage SPP1*delX110lacO64* and imaged at 50 min p.i. PALM images show the super-resolution images reconstructed with 15-nm pixel size. Average of uncertainty from each molecule was used to calculate localization precision. Source images are *Z* projections of the average intensity from raw images. Yellow lines indicate the 2-µm lines perpendicular to the long axis of the cell that were drawn across the DNA viral compartment to plot the intensity profiles shown in B. Scale bars, 500 nm. (B) Intensity line plots across the viral DNA compartment drawn from PALM images (green, LacI-dronPA fusions; magenta, LacI-PAmCherry1 fusions) and from source images (black).

molecule during image acquisition. More molecules were detected, and the surface of the DNA compartment was larger when *LacI* was fused to *bsPAmCherry1* (codon-optimized for *B. subtilis*) compared to when it was fused to the original *PAmCherry1* gene sequence (Fig. 5C). The same trend was observed for *LacI-bsdronPA* localizations and cluster area distributions relative to the non-condon-optimized version *LacI-dronPA*, measured from acquisitions containing an identical number of frames (Fig. 5D). SPP1

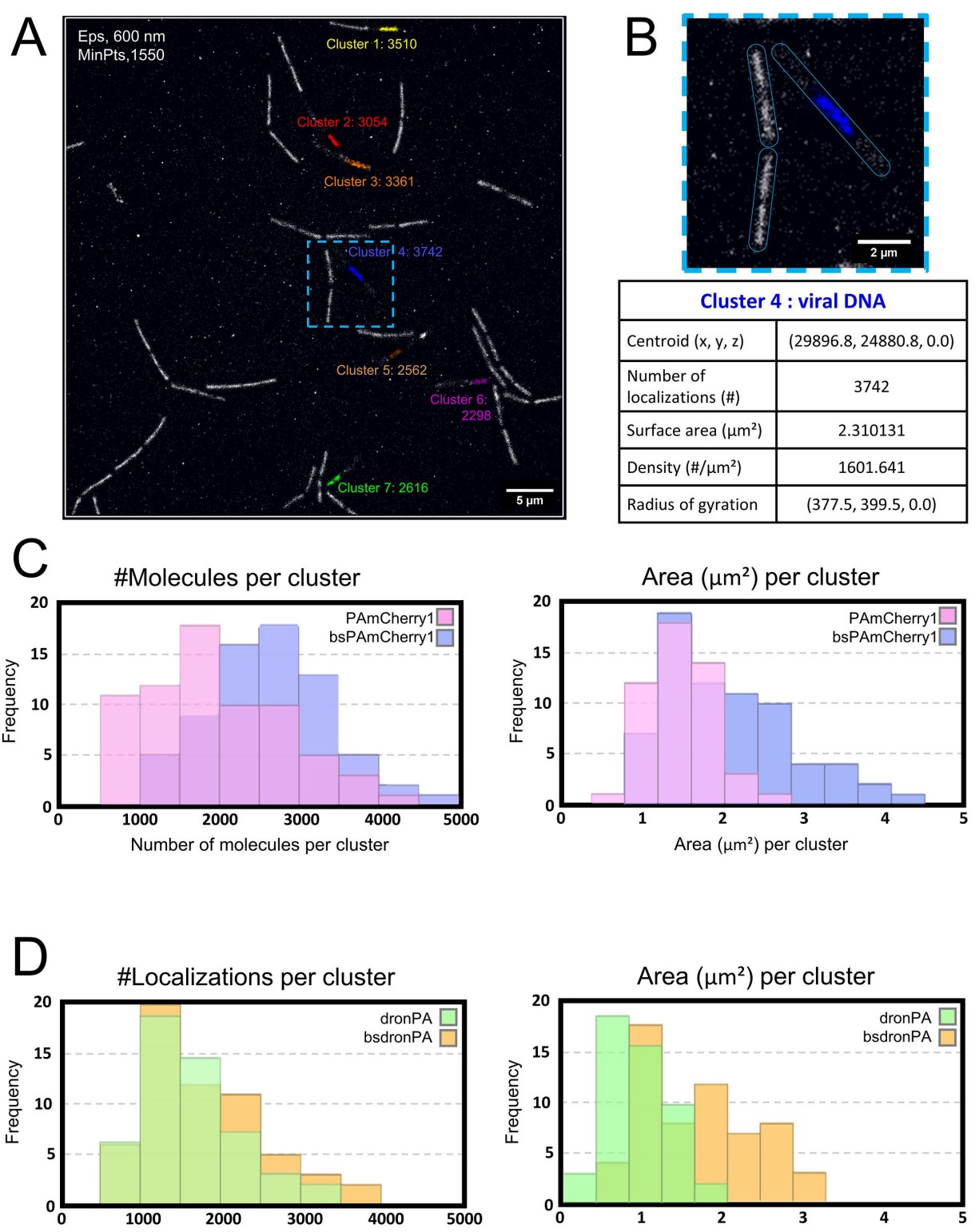

**FIG 5** Cluster analysis of viral DNA. Identification and analysis of viral DNA clusters using DBSCAN cluster analysis. Cells were infected with phage SPP1*delX110lacO64* and imaged on agarose pads 50 min p.i. (A and B) Full field of view (A) of cells (strain RCL0938, LacI-PAmCherry1) used as example to show viral DNA cluster detection and characterization. LacI-FP molecules are soluble and occupy the cytoplasm of non-infected cells. LacI-FP clusters were detected in infected cells and artificially colored. The number of each cluster and the number of localizations detected in that cluster are indicated in the same color. DBSCAN clustering was used with the indicated values for the parameters Epsilon (Eps, refers to the maximum distance two points can be apart from each (Continued on next page)

**Fig 5 (Continued)**

other while belonging to the same cluster) and minimum number of points per cluster (MinPts). The blue dashed square shows the selected Region of interest (ROI) zoomed up in panel B, which contains two non-infected cells and one infected cell containing cluster 4. The cluster information is shown in the table below. (C and D) Number of molecules (LacI-PAmCherry1 fusions, C,) and number of localizations (LacI-dronPA fusions, D) per cluster and surface area (μm²) of the clusters quantified using the DBSCAN algorithm. PAmCherry, $n = 50$ cells; bsPAmCherry, $n = 69$ cells; dronPA, $n = 52$ cells; bsdronPA, $n = 60$ cells.

genome replication is detectable at 10 min p.i. and increases exponentially; at 30 min p.i. one bacterial cell harbors >300 copies of the phage genome (32). Since each viral genome carries ~64 *lacO* sites, we concluded that the number of LacI-FP molecules available may be insufficient for full coverage of all the operator sites contained in the viral DNA compartment. Additionally, codon optimization improved the production of PAmCherry1 and dronPA in *B. subtilis*.

## DISCUSSION

In this report, we describe the construction and testing of a set of eight integration vectors carrying the PALM-compatible dronPA and PAmCherry1 fluorescent proteins, codon-optimized or not for expression in *B. subtilis*. These vectors allow integration of fluorescent fusions at the *amyE* and *thrC* chromosomal loci, expressed under control of the promoters of choice to widen the expression range and induction profiles. They also give the choice of the protein end to which the fluorescent protein can be fused, by single-fragment insertion (N-terminal fusions) or two-fragment insertion (C-terminal fusions) via isothermal assembly. Since PAmCherry1 and dronPA are spectrally different, they can be used for co-localization studies when co-expressed in the same cell. Four of the vectors generated are dedicated to FROS LacI-*lacO* localization studies. Importantly, we show that they can be used for quantitative PALM studies, allowing assessment of the number of molecules (PAmCherry1) or the number of localizations (dronPA). Our results also highlight a limitation of FROS systems when visualizing the temporal and/or spatial organization of multiple DNA copies, such as replicating viral DNA. Cells may not produce enough FP fusions to the repressor protein to achieve full coverage of all operator arrays, which contain numerous copies of the repressor binding site at a specific genomic location. Finally, we show that the PALM-compatible fusions function well in both live and fixed *B. subtilis* cells and that codon optimization of the FP enhanced the production of the fluorescent fusions.

Altogether, our vectors expand the toolbox for SMLM studies in *B. subtilis* and could be also used as templates for genetic engineering. All described plasmids are available through Addgene (IDs 200418, 200419, 200420, 200421, 200423, 400424, 400425, and 400426) and could be of benefit to many researchers in the bacterial community.

## ACKNOWLEDGMENTS

We thank Paulo Tavares and Audrey Labarde for kindly providing strain YB886, plasmid pPT300, and phage SPP1 infection protocols and expertise; Mervenur Tunç for assistance with plasmid and strains handling for the Addgene deposits; Arnaud Chastanet, Magali Ventroux, and Peggy Mervelet for technical suggestions; and Abbelight for providing access to use the Abbelight Neo Analysis software for cluster analysis.

This project was supported by funding from the Agence Nationale de la Recherche (ANR-15-CE11-0010 BacVirRemodel to R.C.-L.) and the European Research Council (ERC) under the Horizon 2020 Research and Innovation program (grant agreement no. 772178, ERC consolidator grant to R.C.-L.).

Conceptualization and validation: I.A. and R.C.-L.; methodology, investigation, and formal analysis: I.A.; resources: R.C.-L.; writing (original draft preparation): I.A; writing (reviewing and editing): I.A. and R.C.-L.; supervision and funding acquisition: R.C.-L.

## AUTHOR AFFILIATION

[1]Université Paris-Saclay, INRAE, AgroParisTech, Micalis Institute, Jouy-en-Josas, France

## PRESENT ADDRESS

Ipek Altinoglu, Abbelight, Cachan, France

## AUTHOR ORCIDs

Ipek Altinoglu ⓘ http://orcid.org/0000-0002-9822-3012
Rut Carballido-Lopez ⓘ http://orcid.org/0000-0001-9383-8811

## FUNDING

| Funder | Grant(s) | Author(s) |
|---|---|---|
| Agence Nationale de la Recherche (ANR) | ANR-15-CE11-0010 | Rut Carballido-Lopez |
| EC \| European Research Council (ERC) | 772178 | Rut Carballido-Lopez |

## AUTHOR CONTRIBUTIONS

Ipek Altinoglu, Conceptualization, Formal analysis, Funding acquisition, Investigation, Methodology, Project administration, Resources, Supervision, Validation, Writing – original draft, Writing – review and editing | Rut Carballido-Lopez, Conceptualization, Formal analysis, Funding acquisition, Investigation, Methodology, Project administration, Resources, Supervision, Validation, Writing – original draft, Writing – review and editing

## DATA AVAILABILITY

The data produced in this study are available upon request from the corresponding author. All described plasmids and their complete sequences are accessible through Addgene.

## ADDITIONAL FILES

The following material is available online.

Open Peer Review

**PEER REVIEW HISTORY (review-history.pdf).** An accounting of the reviewer comments and feedback.

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
