## [Reviewer comments · Microbiology Spectrum]

Microbiology Spectrum

New PALM-compatible integration vectors for use in the Gram-positive model bacterium *Bacillus subtilis*

Ipek Altinoglu and Rut Carballido-Lopez

Corresponding Author(s): Rut Carballido-Lopez, Institut National de Recherche pour l'Agriculture l'Alimentation et l'Environnement Centre Ile-de-France Jouy-en-Josas Antony

Review Timeline:

Submission Date:	July 3, 2024
Editorial Decision:	July 24, 2024
Revision Received:	September 25, 2024
Accepted:	October 3, 2024

Editor: Olga Soutourina

Reviewer(s): Disclosure of reviewer identity is with reference to reviewer comments included in decision letter(s). The following individuals involved in review of your submission have agreed to reveal their identity: Ana Oliveira Paiva (Reviewer #1)

Transaction Report:

DOI: <https://doi.org/10.1128/spectrum.01619-24>

Re: Spectrum01619-24 (New PALM-compatible integration vectors for use in the Gram-positive model bacterium *Bacillus subtilis*)

Dear Dr. Rut Carballido-Lopez:

Thank you for the privilege of reviewing your work. Below you will find my comments, instructions from the Spectrum editorial office, and the reviewer comments.

As you can see both reviewers provided positive comments on your manuscript describing new tools as valuable resources for the community. The manuscript will be accepted after you include requested minor modifications. When revising the manuscript please include some more context in the introduction on the use of fluorescent proteins in *B. subtilis* mentioning associated limitations/drawbacks, and discuss in more depth the results obtained.

Revision Guidelines

Sincerely,
Olga Soutourina
Editor
Microbiology Spectrum

Reviewer #1 (Comments for the Author):

The work by Ipek Altinoglu and Rut Carballido-Lopez sets out to evaluate and optimize fluorescent proteins for PALM imaging in *Bacillus subtilis*. The authors develop integration vectors and describe the use of the fluorescent proteins dronPA and

PAmCherry in PALM microscopy. The authors successfully demonstrate the use of the fluorescent proteins in FROS localization studies. The tools developed will be a valuable resource for the community. I have just minor comments:

- Fig 1B - is redundant, as it its already depicted on Fig 1C.
- Fig 5 AB - missing scale
- Fig 5CD - missing labels for the y, x axis
- Line 378-381 - Can the authors elucidate on their results in comparison with the Labarde A et al (2021)
- Line 380-383 - Please split sentence in two.

Reviewer #2 (Comments for the Author):

This paper describes novel plasmids that can be used to construct fluorescent fusion proteins in *Bacillus subtilis*, that are compatible with PALM superresolution microscopy. The plasmids, sequences etc have been made available to the community via Addgene. This is a useful addition to the B subtilis toolbox that will help many researchers in the community.

I only have a few small comments:

1) A number of items in the description of the plasmid construction is not fully clear:

- The abstract states that cloning sites were added at both ends of the fluorescent protein genes - yet it seems that the genes do have stop codons making it impossible to create a C-terminal fusion, which the current phrasing suggests is possible. This is explained on page 10, but the procedure to clone seems quite laborious so I wonder how often this will be done.
- Line 166 should also mention this was by Gibson assembly
- Fig 1C multiple - not multi - cloning site
- Fig 2A. The position of some of the restriction sites in the map seems not to fit the described cloning procedures and the sequence-based restriction maps on Addgene - ie the SphI and SbfI sites should be upstream from HindIII. Please check carefully (also on addgene, which has the same cartoons).
- Line 168-186: It would be helpful to indicate to the reader that next to the inserted MCS in pIA021/023, additional sites are (re)created, via primer encoding, during the PCR of dronPA and PAmCherry1

2) A recent study (preprint; <https://www.biorxiv.org/content/10.1101/2024.01.04.574214v1>) indicates a potential problem with amyE integrations: {plus minus} 10 % of transformants has a truncation in the gene *ldh* which may affect fermentative metabolism. This defect can only be excluded by chromosome sequencing. For most applications this consideration is moot, nevertheless it would be good to mention this potential effect.

Point-by-point responses to reviewers comments
[manuscript # Spectrum01619-24, by Altinoglu and Carballido-López]

We are thankful to the two reviewers and to the editor, who made nice suggestions to further improve the manuscript. We have revised our manuscript accordingly. Please find below our point-by-point responses to the reviewers and editor comments

Comments by the reviewers are marked in black. Our responses are marked in blue.

Subject: Spectrum01619-24 Decision Letter

Re: Spectrum01619-24 (New PALM-compatible integration vectors for use in the Gram-positive model bacterium *Bacillus subtilis*)

Dear Dr. Rut Carballido-Lopez:

Thank you for the privilege of reviewing your work. Below you will find my comments, instructions from the Spectrum editorial office, and the reviewer comments.

As you can see both reviewers provided positive comments on your manuscript describing new tools as valuable resources for the community. The manuscript will be accepted after you include requested minor modifications. When revising the manuscript please include some more context in the introduction on the use of fluorescent proteins in *B. subtilis* mentioning associated limitations/drawbacks, and discuss in more depth the results obtained.

Thanks for your positive feedback and kindness

Main limitations and drawbacks of the use of FP in *B. subtilis* are the same than in other cellular systems. We mention these limitations in the general introduction (lines 94-104, i.e. expression levels, functionality and stability of FP fusions, overexpression of FP fusions, oligomerization of FPs and use of FP with enhanced folding and maturing kinetics).

We briefly discuss the specific results of SPP1 DNA localisation in the corresponding results section. These results are used as proof-of-concept to show that our integration vectors provide a convenient and versatile workflow of PALM studies in *B. subtilis*. In the general discussion, we therefore prefer remain general and only focus on the generalities and use of our plasmids. We have nevertheless followed the editor's suggestion and added some points in the Discussion section, to highlight some of the general conclusions (see Discussion)

Reviewer #1 (Comments for the Author):

The work by Ipek Altinoglu and Rut Carballido-Lopez sets out to evaluate and optimize fluorescent proteins for PALM imaging in *Bacillus subtilis*. The authors develop integration vectors and describe the use of the fluorescent proteins dronPA and PAmCherry in PALM microscopy. The authors successfully demonstrate the use of the fluorescent proteins in FROS localization studies. The tools developed will be a valuable resource for the community. I have just minor comments:

Thanks to reviewer #1 for their positive overall comment

- Fig 1B - is redundant, as it its already depicted on Fig 1C.

We agree. Thanks for this comment. Panel B has been removed

- Fig 5 AB - missing scale

Thanks for pointing this out. It has been corrected.

- Fig 5CD - missing labels for the y, x axis
Thanks for pointing this out. Corrected.

- Line 378-381 - Can the authors elucidate on their results in comparison with the Labarde A et al (2021)

We believe that the referee meant lines 378-380: "*SPP1 genome replication is detectable at 10 min p.i. and increases exponentially; at 30 min p.i. one bacterial cell harbors > 300 copies of the phage genome [34].*", and that they ask if we could detect a signal at 10 min p.i., with exponentially increase until 30 min p.i. (?)

Our PALM images were taken at a late stage of infection (50 min p.i), when most of the phage genome has been duplicated. We did not do time-course acquisitions throughout infection in these proof-of-concept experiments. Besides, the results from Labarde et al (2021) referred to in these lines relate to qPCR experiments in which bacterial and viral DNA were quantified during infection, from liquid cultures grown in large flasks. It is not possible to make a direct comparison with our data, acquired in different conditions: cells immobilised on agarose pads in closed chambers.

- Line 380-383 - Please split sentence in two.

Done. Original sentence: "Since each viral genome carries ~64 *lacO* sites, we concluded that the number of LacI-FP molecules available may be insufficient for full coverage of all the operator sites contained in the viral DNA compartment, and that codon-optimization improves the production of PAmCherry1 and dronPA in *B. subtilis*."

After splitting in two: "Since each viral genome carries ~64 *lacO* sites, we concluded that the number of LacI-FP molecules available may be insufficient for full coverage of all the operator sites contained in the viral DNA compartment. Additionally, codon-optimisation improved the production of PAmCherry1 and dronPA in *B. subtilis*."

Reviewer #2 (Comments for the Author):

This paper describes novel plasmids that can be used to construct fluorescent fusion proteins in *Bacillus subtilis*, that are compatible with PALM superresolution microscopy. The plasmids, sequences etc have been made available to the community via Addgene. This is a useful addition to the *B. subtilis* toolbox that will help many researchers in the community.

Thanks to reviewer #2 too for their positive feedback

I only have a few small comments:

1) A number of items in the description of the plasmid construction is not fully clear:

- The abstract states that cloning sites were added at both ends of the fluorescent protein genes - yet it seems that the genes do have stop codons making it impossible to create a C-terminal fusion, which the current phrasing suggests is possible. This is explained on page 10, but the procedure to clone seems quite laborious so I wonder how often this will be done.

We understand the concern of the referee regarding C-terminal fusions. Our system offers rapid single fragment insertion via isothermal assembly for N-Terminal fusions while it requires 2-fragments (one full backbone annealing without stop codon of the FP and a second fragment with the gene of interest, with complementary primers) for C-terminal fusions. This was explained in lines 310-314 (original manuscript line numbers). Nowadays, 2-fragments isothermal assembly is an accessible and straightforward approach for cloning, which in our case makes C-terminal fusions readily available. We have added a sentence indicating this in line 315, and also a sentence in the discussion to point to the reader the one-fragment (N-terminal fusions) vs two-fragments (C-terminal fusions) isothermal assembly to be performed.

- Line 166 should also mention this was by Gibson assembly

Done. Thanks for this useful comment

Original text: “*amplified with primers oIA003 and oIA004 was cloned into pDG1664 [23] (thrC::erm2) digested (...)*”

Corrected text: “*amplified with primers oIA003 and oIA004 was cloned by Gibson assembly into pDG1664 [23] (thrC::erm2) digested (...)*”

- Fig 1C multiple - not multi - cloning site

Corrected (panel shown as Fig 1B now as per reviewer #1 suggestion). Thanks. Sorry for the typo.

- Fig 2A. The position of some of the restriction sites in the map seems not to fit the described cloning procedures and the sequence-based restriction maps on Addgene - ie the SphI and SbfI sites should be upstream from HindIII. Please check carefully (also on addgene, which has the same cartoons).

Thanks for spotting this! It has been corrected by placing SphI and SbfI upstream HindIII in Figure 2A and also in the Addgene files. All the other restriction sites have been double-checked for all plasmids, no further mistakes were found.

- Line 168-186: It would be helpful to indicate to the reader that next to the inserted MCS in pIA021/023, additional sites are (re)created, via primer encoding, during the PCR of dronPA and PAmCherry1

Done. Thanks for this useful suggestion. The following sentence has been added in line 183 > “The newly constructed plasmids (pIA027, pIA028, pIA034 and pIA033) re-created two additional restriction sites, EagI and NotI, next to the MCS via primer encoding”

2) A recent study (preprint; <https://www.biorxiv.org/content/10.1101/2024.01.04.574214v1>) indicates a potential problem with amyE integrations: {plus minus} 10 % of transformants has a truncation in the gene *ldh* which may affect fermentative metabolism. This defect can only be excluded by chromosome sequencing. For most applications this consideration is moot, nevertheless it would be good to mention this potential effect.

Thanks for pointing this out. In this preprint by Dierksheide and Li, a discontinuous *amyE*-back homology region was specifically detected in plasmid pDR111 and its parental plasmid, the original *amyE* double-crossover integration vector pBGtrp. In our study, we used pDG364, a derivative of the transcriptional fusion vector pDG268 (Antoniewski et al., 1990), which is itself a derivative of the pDH32 vector (referred to Shimotsu and Henner, 1986 in the literature, but we did not find any reference to this plasmid in Shimotsu and Henner 1986). Unless we missed something, the literature does not allow to track the origin of plasmid pDH32 with certainty, and thus to determine if it is a derivative of pBGtrp or not. Other references to pDH32 in the literature include: “*The single-copy transcriptional fusion vector pDH32 (D. Henner, personal communication) (...) is similar to the ptrpBG1 translational fusion vector of Shimotsu and Henner (36)*” (Boylan et al, J. Bacteriol 1989) or ‘*pDH32 (provided by Dennis Henner)*’ (Grandoni et al, J Bacteriol 1993), etc.

In all instances, the potential impact of a possible gap in the homology region (if inherited from the original *amyE* integration vector pBGtrp) resulting in a loss-of-function of the downstream *ldh* gene (encoding lactate dehydrogenase) in 10% of the transformants, would not affect our results and is irrelevant to most applications in which our plasmids will be used. Since this BioRxiv preprint has not yet undergone peer-review and we are uncertain if this applies to our *amyE* parental plasmid pDG364, we prefer not to mention the preprint at this stage to prevent possible confusions.

Re: Spectrum01619-24R1 (New PALM-compatible integration vectors for use in the Gram-positive model bacterium *Bacillus subtilis*)

Dear Dr. Rut Carballido-Lopez:

Thank you for the revision of the article according to the reviewers recommendations.

I am pleased to inform you that your manuscript has been accepted, and I am forwarding it to the ASM production staff for publication. Your paper will first be checked to make sure all elements meet the technical requirements. ASM staff will contact you if anything needs to be revised before copyediting and production can begin. Otherwise, you will be notified when your proofs are ready to be viewed.

Sincerely,
Olga Soutourina
Editor
Microbiology Spectrum